behaviour/health and disease and epidemiology/ecology

avian, eco-immunology, migration, recovery, longitudinal (within-individual) data, immunity

**Author for correspondence:**
Cas Eikenaar
e-mail: cas.eikenaar@ifv-vogelwarte.de

# Migrating birds rapidly increase constitutive immune function during stopover

Cas Eikenaar[1], Sven Hessler[1] and Arne Hegemann[2]

[1]Institute of Avian Research, An der Vogelwarte 21, Wilhelmshaven 26386, Germany
[2]Department of Biology, Lund University, Ecology Building, Lund 223 62, Sweden

 CE, 0000-0003-1049-8036; AH, 0000-0002-3309-9866

Migratory flight is physiologically highly demanding and has been shown to negatively affect multiple parameters of constitutive immune function (CIF), an animal's first line of physiological defence against infections. In between migratory flights, most birds make stopovers, periods during which they accumulate fuel for the next flight(s). Stopovers are also commonly thought of as periods of rest and recovery, but what this encompasses is largely undefined. Here, we show that during stopover, northern wheatears *Oenanthe oenanthe*, a long-distance migratory bird, can rapidly increase constitutive innate immune function. We caught and temporarily caged birds under ad libitum food conditions at a stopover site in autumn. Within 2 days, most birds significantly increased complement activity and their ability to kill microbes. Changes in immune function were not related to the birds' food intake or extent of fuel accumulation. Our study suggests that stopovers may not only be important to refuel but also to restore immune function. Additionally, the increase in CIF could help migrating birds to deal with novel pathogens they may encounter at stopover sites.

## 1. Introduction

The immune system helps to keep an organism healthy, which is not only important for survival but also for individual performance during the various annual-cycle stages. For example, experimentally mimicked bacterial infection reduces offspring feeding rate and reproductive success in house sparrows (*Passer domesticus*) [1]. Mimicked bacterial infections also prolong stopovers, the periods of fuel accumulation in between migratory flights, and change stopover behaviour in migrating birds [2]. This latter result suggests that getting infected during stopover could slow down migration, which may have negative carry-over effects into the breeding and wintering seasons [3,4].

An animal's immune system includes both constitutive (baseline) defences, the always present immediate first lines of defence, and induced immune responses [5,6]. The main benefit of an improved constitutive immune function (CIF) is probably increased likelihood of clearing pathogens before they start replicating and establishing themselves inside a host, while an innate immune response is mounted when a pathogen starts replicating and establishing itself in the body. Baseline immune function and immune responses are differently regulated and baseline immune function often does not correlate with the strength of an immune response [7,8]. While immune responses are important to ensure survival, immune responses are energetically costly [9,10], always include self-damage, and often include 'opportunity costs', i.e. lost opportunities in life history, resulting in fitness costs [10–13]. Hence, a stronger CIF (baseline) should be favoured in order to avoid mounting an actual immune response.

As stopovers usually last only a few days, migrants at stopover may be particularly keen on avoiding immune responses and hence rely heavily on CIF [14]. However, migratory endurance flight possibly weakens birds' CIF; although intense flight during races does not affect CIF in non-migratory racing pigeons (*Columba livia*) [15], wind-tunnel studies on migrant bird species show that flight can negatively impact CIF [16,17]. In northern wheatears (*Oenanthe oenanthe*) caught at stopover, several parameters of CIF were positively correlated with the birds' fuel stores, which may also suggest that migratory flight negatively affects CIF [18]. As a weak first line of immune defence probably increases the severity of infection, and consequently may decrease survival [19,20], migrants may be expected to boost their CIF while at stopover, especially when arriving in a poor immunological state and/or at stopover sites where high densities of other birds promote pathogen transmission. A hint that this indeed happens comes from a cross-sectional study on red knots (*Calidris canutus*) showing that CIF is stronger in individuals storing fat than in individuals recovering protein, and thus probably stronger in individuals that have been at stopover longer [14]. However, to our knowledge, no study has directly investigated whether individual migrants are able to boost their CIF while at stopover, probably because it is notoriously difficult to re-trap migrants during their stopovers to collect the necessary longitudinal data. Temporary caging of migrants at stopover arguably is the next best alternative; however, in such set-ups, many bird species show signs of physiological stress, which can negatively affect immune defences [21–23]. Here, we studied changes in CIF in temporarily caged migrating northern wheatears (wheatear hereafter) during autumn migration. Importantly, migrating wheatears exhibit none of the common signs of ongoing stress when temporarily caged. Upon caging, they almost immediately start eating any provided food, and their corticosterone levels return to baseline within a few hours [24]. We captured wheatears at stopover and held them in captivity for 3 days (matching stopover durations of this species [25,26]) and assessed whether they can boost CIF during this time. Each bird was blood-sampled around noon on both the first and third full day in captivity for the measurement of three parameters of CIF: microbial killing capacity against *Escherichia coli*, complement activity (lysis titres) and natural antibodies (agglutination titres). Natural antibodies circulate in the blood without previous exposure to a particular antigen and can recognize and neutralize pathogens directly or indirectly by activation of the complement cascade, which ends in cell lysis [27].

# 2. Methods

## 2.1. Data collection

In September 2018, during autumn migration, wheatears were caught on Helgoland (54°11′ N, 07°55′ E), a small island *ca* 50 km off the German North Sea coastline. After capture, birds (23 first year and eight adult birds, aged after [28]) were ringed, wing length (maximum chord after [28]) was measured to the nearest 0.5 mm and body mass was measured to the nearest 0.1 g. The birds were then placed individually into cages (40 × 40 × 30 cm), which were set up in a room with constant temperature (approx. 20°C) and artificial lighting (14 L : 10 D, following [29,30]). The birds had ad libitum access to water. Upon caging and each subsequent morning at lights-on, a food tray with 40 g of mealworms (approx. 200 mealworms) was placed in each cage. Food trays were removed at lights-off or at release (see below), and the amount of food (g) eaten that day was recorded. Each morning at lights-on, i.e. when the birds had an empty gastrointestinal tract, the body mass of all birds was measured to the nearest 0.1 g. Each bird was blood-sampled (*ca* 70 µl) on both the first and third full day in captivity, from the right and left wing vein, respectively. All blood samples were taken close to 12.00 noon local time, within 10 min from entering the room. Plasma was separated immediately after blood-sampling

and stored, first at −20°C and later at −50°C until assaying. After the second blood-sampling, birds were released. Red blood cells were used for molecular sexing, which showed that of our birds, 14 were female and 17 were male.

Wing length was used to estimate lean body mass (LBM), following [31]. The estimates of LBM were used to calculate fuel stores: (body mass (g) − LBM (g))/LBM (g), following [31]. An estimate of fuel stores thus represents the amount of fuel (both fat tissue and proteins) a bird carries relative to its lean body mass. Negative fuel loads may occur when tissue not included in fat and flight muscle scores (used in the calculation of LBM), e.g. non-visible (endogenous) fat and/or protein from other muscles than the flight muscle, is being catabolized.

## 2.2. Immune assays

We quantified the microbial killing capacity (against *E. coli*) of plasma following the method described by [32] with a few modifications. We used 3 µl of plasma and mixed it in 4 µl of $10^5$ *E. coli* solution. Plates were incubated at 37°C for 12 h, subsequently vortexed for 1 min at 100 rpm, and read at 600 nm using a microplate reader [33]. We calculated the per cent of *E. coli* killed relative to the growth of *E. coli* in wells not containing plasma, following [32]. We used four negative controls per plate to ensure that there was no contamination.

We quantified complement activity and natural antibody titres using a haemolysis–haemagglutination assay [34]. In brief, red blood cells from rabbits (Envigo RMS Ltd, UK) were incubated in serially diluted plasma samples. We used assay plate images taken 20 min after incubation to score agglutination and images made 90 min after incubation to score lysis. All images were randomized and each serial dilution was scored twice blindly with respect to sample identity. Half scores between two titres were given when the termination of lysis or agglutination was ambiguous. When the two scores of either lysis or agglutination were less than 1 titre apart their mean was used in the analyses. When two scores were greater than or equal to 1 titre apart the sample was scored a third time and the median was used. Owing to limited plasma volumes, we used 10 µl of plasma (instead of 25 µl). For two individuals, we did not have enough plasma to run this assay. In both assays, the two repeated samples of an individual were always positioned next to each other on the same plate.

## 3. Results

In most individuals, microbial killing capacity and lysis increased from the first to third day in captivity (Wilcoxon signed-ranks test: $Z = -3.27$, $p = 0.001$, $n = 31$ and $Z = -3.78$, $p < 0.001$, $n = 29$, respectively; figure 1). Agglutination did not consistently change, with some individuals showing an increase and others a decrease (Wilcoxon signed-ranks test: $Z = -0.26$, $p = 0.80$, $n = 29$; figure 1). The magnitude of changes in the three parameters of CIF during captivity were, or tended to be, positively correlated with each other (table 1).

The total amount of food consumed from the first full day until release was very high (mean ± s.d. = 58.6 g ± 10.6, range 32.2–81.6 g, $n = 31$), equalling roughly 2.5 times the birds' lean body mass. This high food intake resulted in most birds accumulating a considerable amount of fuel from the first to third morning in captivity (mean ± s.d. = 0.15 ± 0.07, range −0.04 to 0.27, $n = 31$). Neither total food intake nor the change in fuel stores were correlated with changes in parameters of CIF (table 1). Sex did not explain changes in microbial killing capacity (Mann–Whitney *U*-test: $Z = 1.27$, $p = 0.22$, 17 males, 14 females), lysis (Mann–Whitney *U*-test: $Z = 0.29$, $p = 0.78$, 16 males, 13 females) or agglutination (Mann–Whitney *U*-test: $Z = 0.02$, $p = 0.98$, 16 males, 13 females). The limited number of adult birds in our sample precluded testing for an age effect on the changes in CIF.

## 4. Discussion

In migrating birds, the rate of fuel expenditure in flight is higher than the rate of fuel accumulation at stopover [35]. Therefore, the total duration of migration is primarily dictated by stopover durations and to a lesser extent by the time spent flying [36,37]. Understanding stopover ecology thus is crucial to understanding bird migration. Stopovers serve to accumulate fuel, but are also typically thought to allow migrants to 'rest' and recover from migratory endurance flight (e.g. [38–45]), even though within-individual data to substantiate this hypothesis have, until now, been lacking.

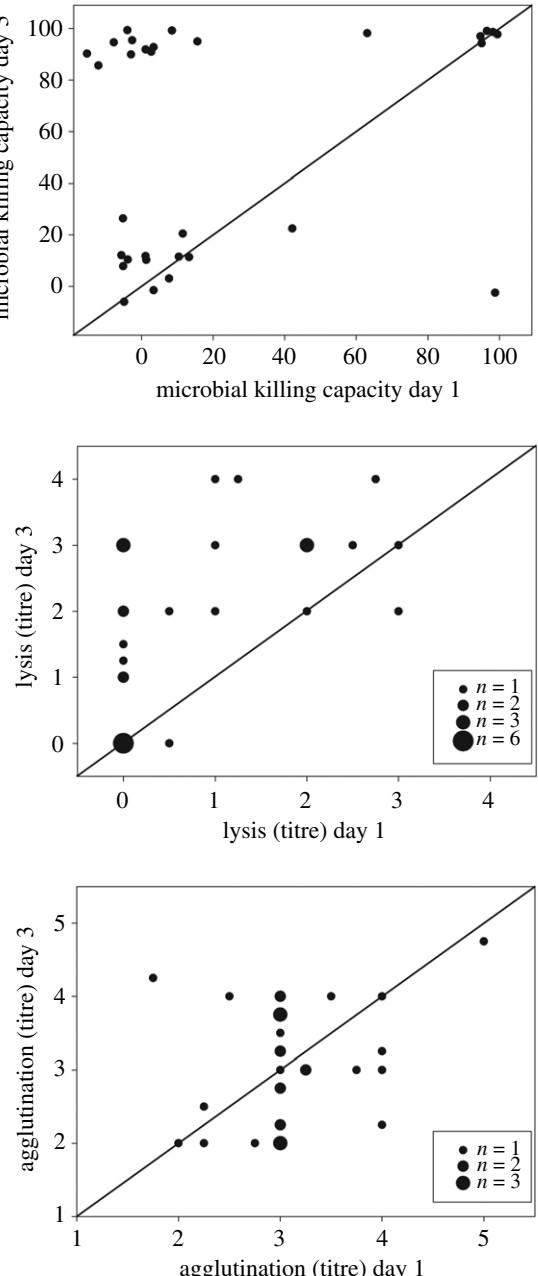

**Figure 1.** Three parameters of constitutive immune function in migrating northern wheatears sampled on both the first and third full day of captivity at a stopover site (day 1 and day 3, respectively). $n = 31$ for microbial killing capacity and $n = 29$ for lysis and agglutination. Symbol sizes reflect the number of individual birds whose scores were identical. Values above the line indicate that birds increased immune function, while values below the line indicate that individuals decreased immune function from day 1 to day 3.

Cross-sectional studies did hint that migrants may need stopovers to recover from the physiological challenges of migratory flight; individuals having spent much time at stopover had lower oxidative damage to lipids, and stronger CIF and immune responses than individuals having spent little time at stopover [14,42,46]. In our longitudinal study, we now show that migrating wheatears temporarily caged at a stopover site under ad libitum food conditions are able to increase, often substantially, parameters of CIF within just 2 days. This increase in CIF strengthens the idea that indeed migrants can use stopovers not only to refuel but also to recover physiological systems. Our study cannot disentangle though, the relative importance of stopovers for refuelling and recovery of CIF (or other physiological systems). The finding that parameters of CIF appear to have little effect on the likelihood of departure from a stopover site [13,47] suggests that in migrating passerines, the process of recovering CIF might not determine the length of stopovers.

**Table 1.** Spearman's correlation coefficients for changes in three parameters of constitutive immune function, the change in fuel stores and total food intake in migrating northern wheatears (*O. oenanthe*) temporarily caged at a stopover site. (Microbial: microbial killing capacity. $^{*}p < 0.05$; $^{**}0.05 < p < 0.10$.)

| variable | microbial | lysis (titre) | agglutination (titre) | fuel stores | food intake |
|---|---|---|---|---|---|
| microbial | — | | | | |
| lysis | 0.384* | — | | | |
| agglutination | 0.361** | 0.342** | — | | |
| fuel stores | 0.017 | 0.017 | −0.269 | — | |
| food intake | −0.08 | −0.001 | −0.137 | 0.679* | — |

The increase in CIF could also function to enhance the birds' ability to fight off infections while at stopover [14]. This may be particularly relevant for wheatears, which during their extremely long migratory journeys visit multiple, often widely separated stopover sites [37,48] that could hold pathogens that they have never encountered before. The increase in immune function we observed in our study also highlights the immune system's sensitivity to changing conditions [49–52], even in very short timeframes. Interestingly, for many individuals, the increase in microbial killing capacity appears stronger than the increase in lysis. This is probably because the bacteria killing capacity is a broad, integrative measure of innate immune function [32,53,54]. By contrast, complement activity (lysis titre) and natural antibody titres (agglutination titre) reflect more tailored parameters ([34], and references therein). Our data suggest that some parameters (captured in the microbial killing capacity) might be upregulated to a stronger degree in short timeframes than complement activity and natural antibodies titres, respectively.

The reason why certain individuals with a relatively low initial CIF strongly increased (multiple) parameters of CIF, while others did not or to a lesser extent, remains unclear. Perhaps some individuals need more time (i.e. longer stopovers) to be able to do so. This would match the observation that whereas most wheatears make short, 1–3 day stopovers on Helgoland, some individuals stay considerably longer, sometimes more than a week [25,26]. The among-individual variation in boosting CIF, present in both sexes, at least did not result from differences in food intake or fuel accumulation, even though there was substantial variation in these processes among birds. This may suggest that upregulating CIF occurs independently from the amount of available food; however, we cannot exclude the possibility that the 'mild' caging conditions (20°C and ad libitum food) masked a potential relationship between food availability and the ability to increase CIF. Still, the ad libitum food conditions in our study are not particularly unnatural: in wheatears, the rate of fuel accumulation in the field is apparently limited by metabolic capacity and not by food availability [55]. Furthermore, fuel accumulation is similar in free-flying individuals and caged individuals kept under the same conditions as used in the current study [56]. Hence, combining the increase in CIF reported from cross-sectional field data [14] with the data presented here, we hypothesize that free-living wheatears (and other migrants) without unlimited food supply are also able to increase CIF during stopover. Collecting longitudinal immune data on free-flying migrants to test this is a next major challenge. With the current study, we have nonetheless placed an important piece of the puzzle of how birds deal with the physiological challenges they face during migration.

Ethics. All procedures were approved by the Ministry of Energy, Agriculture, the Environment, Nature and Digitalization, Schleswig-Holstein, Germany (permit number V 242-37068/2016).

Data accessibility. Data are available as the electronic supplementary material.

Authors' contributions. C.E. conceived of the study, S.H. and C.E. collected data and C.E. performed statistical analyses. A.H. performed all laboratory analyses. C.E. and A.H. together wrote the article. All authors approved of the current version, and agree to be accountable for its contents.

Competing interests. We have no competing interests.

Funding. The study was supported with grants from the Deutsche Forschungsgemeinschaft (DFG) awarded to C.E. (grant no. EI 1048/3-1), and from the Swedish Research Council awarded to A.H. (grant no. 2018-04278).

Acknowledgements. We thank Jana Schäfer for help with data collection, Jochen Dierschke and Klaus Müller for logistic support on Helgoland and Kevin Matson, Franz Bairlein and three anonymous reviewers for useful suggestions on the manuscript.

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
