## [Reviewer comments · Royal Society Open Science]

Review History

RSOS-192031.R0 (Original submission)

Review form: Reviewer 1 (Camila Gómez)

Is the manuscript scientifically sound in its present form?

Yes

Are the interpretations and conclusions justified by the results?

Yes

Is the language acceptable?

Yes

Do you have any ethical concerns with this paper?

No

Have you any concerns about statistical analyses in this paper?

No

Recommendation?

Accept with minor revision (please list in comments)

Comments to the Author(s)

This manuscript presents very interesting data on recovery and activation of the immune system in a migratory species on stopover. They have addressed well the previous reviewer's comments and their manuscript has improved even more as a result.

I only had very minor observations that authors can address easily.

L44 - Why (novel) in parenthesis?

L70, 77, 85, 177 - Perhaps these lines should be the start of new paragraphs? It might be that something happened when formatting the proofs, but both the introduction and the discussion seem to be one single huge paragraph. Please check and make sure that the text is appropriately formatted with paragraphs.

L76 - Please check whether the expression 'birds at stopover' should be changed to 'birds on stopover'. This is repeated in various sections throughout the manuscript and I believe that most literature uses 'on stopover'. I may be wrong.

L161 - Please add a short sentence saying that you could not test for age effects.

Supplementary data - Please add a column for sex and one for age in the supplementary material.

Review form: Reviewer 2 (Nicholas P. C. Horrocks)

Is the manuscript scientifically sound in its present form?

Yes

Are the interpretations and conclusions justified by the results?

Yes

Is the language acceptable?

Yes

Do you have any ethical concerns with this paper?

No

Have you any concerns about statistical analyses in this paper?

No

Recommendation?

Accept with minor revision (please list in comments)

Comments to the Author(s)

This study describes the ability of a long-distance migrant to rapidly increase constitutive innate immunity during a stopover with ad libitum food. The study is well written and clearly laid out. It adds to our knowledge of the role that stopovers play in the ecology and physiology of migrating birds as well as emphasising how immune function is a dynamic process that can change over quite short time periods.

I have been asked to review this manuscript for a second time (previously for Biology Letters), so I only have minor comments that relate to improving the clarity, and perhaps making use of the fact that the longer format of RSOS compared to Biology Letters allows for a more comprehensive presentation and discussion of the results. Thank you for addressing so thoroughly all the comments and suggestions that I made in my previous review of your manuscript. I hope you find these additional comments useful.

L37 - Change to "Here, we show that..."

L74-L77 - I was confused by the terminology used here. Constitutive immunity often is inflammatory, so this doesn't quite seem to make sense. Perhaps, within the innate immune system you are making the distinction between constitutive innate immunity and induced innate defences (see Lee 2006 Linking immune defenses and life history at the levels of the individual and the species. *Integ. Comp. Biol.* 46: 1000-1015)? If so, this needs to be made clear.

At L76 you say that stopover migrants might prefer to use CIF because they are keen on avoiding immune responses, but CIF is a form of immune response/defence. I think you mean that stopover migrants might not want to use induced or adaptive immune responses, because these take longer to develop, and in the case of induced innate immune defences, are energetically costly?

L78 - The context of racing with pigeons might be a bit lost to the reader here, unless they are familiar with the paper. Consider changing to something like "...although intensive flight during races does not affect CIF in non-migratory racing pigeons..."

L92 - Consider changing to "...can negatively affect immune defences."

L96 - Perhaps consider adding in a sentence here to make it clear how long you held birds for. Otherwise it feels a little like you dive straight into the methods. Something like "We captured migrating wheatears at stopover and held them for three days in captivity (which mimics the average stop-over time of this species at our study site), to assess whether they could boost CIF during this time. Each bird was blood-sampled..." Currently the text reads a bit like you are still in *Biology Letters* format, with minimal space for providing more thorough details. I think you can afford to include a few more words now. I also think it is worth to state upfront that the three days in captivity matches a typical stop-over period for this species. I realise that you do this at L187-190 (and should continue to do so - thanks for making that change from the previous version), but I think including it in brief detail early on helps reassure the reader that your study has direct biological relevance to stopover behaviour in this species.

L101 - I think it would be beneficial to finish with some predictions here, just to remind the reader what it is you are aiming to test with your study. Something along the lines of "If migratory wheatears are able to boost CIF during stop-over, then this predicts that CIF should be higher at day three than at day one, whereas if wheatears are unable to boost CIF, we would expect no difference in levels between the start and end of captivity."

L106 - Change to "After capture, birds..."

L124-126 - Thanks for now including this clear explanation.

L169 - Change to "...to substantiate this hypothesis have, until now, been lacking."

L171-172 - Again, I would encourage greater clarity in terminology here. What do you mean by "better CIF and immune responses"? CIF is a type of immune response.

L181-182 - This point about encountering novel pathogens seems particularly relevant to young birds that have not previously migrated. Given that your sample included mostly first year birds, do you think you would have found similar results if you had only captured adult birds?

Your sample sizes may not permit it, but I wondered if it might be informative to test for an age effect (first year vs adult birds) on change in immune parameters? Perhaps older birds are more experienced which might affect their flight performance and thus physical state when captured, which could in turn affect ability to upregulate CIF? The fact that you found no effect of food

intake or changes in fuel stores would suggest this is not the case, but it could be interesting to check, just in case.

Figure 1 – This figure really clearly illustrates your results. One striking thing is the result for microbial killing capacity. You can almost completely split the data here into two groups. Those birds that had 100% killing ability at day 1 maintained it at day 3. Perhaps these individuals can be considered as ‘high quality’ in some way? For most other individuals, killing capacity at day 1 was very low – less than 20%. Yet at day 3 it had shot up to greater than 80%. This seems like a dramatic change, and worthy of more discussion, especially when contrasted with the results for lysis and agglutination, where there was much more variation. Is there something specific about microbial killing capacity that makes it more important to up-regulate than lysis or agglutination? Or is this a cheaper form of CIF, so can be more easily up-regulated? Given that you are now not word count-limited, you might consider elaborating on your results a bit more in the discussion.

Supplementary data – Please include sex and age data in your Excel file.

Decision letter (RSOS-192031.R0)

17-Dec-2019

Dear Dr Eikenaar

On behalf of the Editors, I am pleased to inform you that your Manuscript RSOS-192031 entitled "Migrating birds rapidly increase constitutive immune function during stopover" has been accepted for publication in Royal Society Open Science subject to minor revision in accordance with the referee suggestions. Please find the referees' comments at the end of this email.

The reviewers and handling editors have recommended publication, but also suggest some minor revisions to your manuscript. Therefore, I invite you to respond to the comments and revise your manuscript.

- Ethics statement

- Data accessibility

<http://datadryad.org/submit?journalID=RSOS&manu=RSOS-192031>

- **Competing interests**

- **Authors' contributions**

- **Acknowledgements**

- **Funding statement**

Because the schedule for publication is very tight, it is a condition of publication that you submit the revised version of your manuscript before 26-Dec-2019. Please note that the revision deadline will expire at 00.00am on this date. If you do not think you will be able to meet this date please let me know immediately.

If your manuscript is newly submitted and subsequently accepted for publication, you will be asked to pay the article processing charge, unless you request a waiver and this is approved by Royal Society Publishing. You can find out more about the charges at <https://royalsocietypublishing.org/rsos/charges>. Should you have any queries, please contact openscience@royalsociety.org.

on behalf of Prof Kevin Padian (Subject Editor)
openscience@royalsociety.org

Associate Editor Comments to Author:

The reviewers are largely pleased with the changes you have implemented following prior review at one of our sister journals; however, a few recommendations have been made in this version that we'd like you to take into consideration.

Reviewer comments to Author:

Reviewer: 1

Comments to the Author(s)

This manuscript presents very interesting data on recovery and activation of the immune system in a migratory species on stopover. They have addressed well the previous reviewer's comments and their manuscript has improved even more as a result.

I only had very minor observations that authors can address easily.

L44 - Why (novel) in parenthesis?

L70, 77, 85, 177 - Perhaps these lines should be the start of new paragraphs? It might be that something happened when formatting the proofs, but both the introduction and the discussion seem to be one single huge paragraph. Please check and make sure that the text is appropriately formatted with paragraphs.

L76 - Please check whether the expression 'birds at stopover' should be changed to 'birds on stopover'. This is repeated in various sections throughout the manuscript and I believe that most literature uses 'on stopover'. I may be wrong.

L161 - Please add a short sentence saying that you could not test for age effects.

Supplementary data - Please add a column for sex and one for age in the supplementary material.

Reviewer: 2

Comments to the Author(s)

This study describes the ability of a long-distance migrant to rapidly increase constitutive innate immunity during a stopover with ad libitum food. The study is well written and clearly laid out. It adds to our knowledge of the role that stopovers play in the ecology and physiology of migrating birds as well as emphasising how immune function is a dynamic process that can change over quite short time periods.

I have been asked to review this manuscript for a second time (previously for *Biology Letters*), so I only have minor comments that relate to improving the clarity, and perhaps making use of the fact that the longer format of *RSOS* compared to *Biology Letters* allows for a more comprehensive presentation and discussion of the results. Thank you for addressing so thoroughly all the comments and suggestions that I made in my previous review of your manuscript. I hope you find these additional comments useful.

L37 - Change to "Here, we show that..."

L74-L77 - I was confused by the terminology used here. Constitutive immunity often is inflammatory, so this doesn't quite seem to make sense. Perhaps, within the innate immune system you are making the distinction between constitutive innate immunity and induced innate defences (see Lee 2006 Linking immune defenses and life history at the levels of the individual and the species. *Integ. Comp. Biol.* 46: 1000-1015)? If so, this needs to be made clear.

At L76 you say that stopover migrants might prefer to use CIF because they are keen on avoiding immune responses, but CIF is a form of immune response/defence. I think you mean that stopover migrants might not want to use induced or adaptive immune responses, because these take longer to develop, and in the case of induced innate immune defences, are energetically costly?

L78 - The context of racing with pigeons might be a bit lost to the reader here, unless they are familiar with the paper. Consider changing to something like "...although intensive flight during races does not affect CIF in non-migratory racing pigeons..."

L92 – Consider changing to “...can negatively affect immune defences.”

L96 – Perhaps consider adding in a sentence here to make it clear how long you held birds for. Otherwise it feels a little like you dive straight into the methods. Something like “We captured migrating wheatears at stopover and held them for three days in captivity (which mimics the average stop-over time of this species at our study site), to assess whether they could boost CIF during this time. Each bird was blood-sampled...” Currently the text reads a bit like you are still in Biology Letters format, with minimal space for providing more thorough details. I think you can afford to include a few more words now. I also think it is worth to state upfront that the three days in captivity matches a typical stop-over period for this species. I realise that you do this at L187-190 (and should continue to do so – thanks for making that change from the previous version), but I think including it in brief detail early on helps reassure the reader that your study has direct biological relevance to stopover behaviour in this species.

L101 – I think it would be beneficial to finish with some predictions here, just to remind the reader what it is you are aiming to test with your study. Something along the lines of “If migratory wheatears are able to boost CIF during stop-over, then this predicts that CIF should be higher at day three than at day one, whereas if wheatears are unable to boost CIF, we would expect no difference in levels between the start and end of captivity.”

L106 – Change to “After capture, birds...”

L124-126 – Thanks for now including this clear explanation.

L169 – Change to “...to substantiate this hypothesis have, until now, been lacking.”

L171-172 – Again, I would encourage greater clarity in terminology here. What do you mean by “better CIF and immune responses”? CIF is a type of immune response.

L181-182 – This point about encountering novel pathogens seems particularly relevant to young birds that have not previously migrated. Given that your sample included mostly first year birds, do you think you would have found similar results if you had only captured adult birds?

Your sample sizes may not permit it, but I wondered if it might be informative to test for an age effect (first year vs adult birds) on change in immune parameters? Perhaps older birds are more experienced which might affect their flight performance and thus physical state when captured, which could in turn affect ability to upregulate CIF? The fact that you found no effect of food intake or changes in fuel stores would suggest this is not the case, but it could be interesting to check, just in case.

Figure 1 – This figure really clearly illustrates your results. One striking thing is the result for microbial killing capacity. You can almost completely split the data here into two groups. Those birds that had 100% killing ability at day 1 maintained it at day 3. Perhaps these individuals can be considered as ‘high quality’ in some way? For most other individuals, killing capacity at day 1 was very low – less than 20%. Yet at day 3 it had shot up to greater than 80%. This seems like a dramatic change, and worthy of more discussion, especially when contrasted with the results for lysis and agglutination, where there was much more variation. Is there something specific about microbial killing capacity that makes it more important to up-regulate than lysis or agglutination? Or is this a cheaper form of CIF, so can be more easily up-regulated? Given that you are now not word count-limited, you might consider elaborating on your results a bit more in the discussion.

Supplementary data – Please include sex and age data in your Excel file.

Author's Response to Decision Letter for (RSOS-192031.R0)

See Appendix A.

Decision letter (RSOS-192031.R1)

06-Jan-2020

Dear Dr Eikenaar,

It is a pleasure to accept your manuscript entitled "Migrating birds rapidly increase constitutive immune function during stopover" in its current form for publication in Royal Society Open Science. The comments of the reviewer(s) who reviewed your manuscript are included at the foot of this letter.

on behalf of Mr Andrew Dunn (Associate Editor) and Kevin Padian (Subject Editor)
openscience@royalsociety.org

Associate Editor Comments to Author (Mr Andrew Dunn):

Associate Editor: 1

Comments to the Author:

(There are no comments.)

Reviewer comments to Author:

Appendix A

Dear Editor,

Thank you very much for your positive letter. We were glad to see that the reviewers were content with the changes we made in response to their comments, and for their last few minor comments, all of which we have incorporated in the manuscript.

For clarity we have pasted the comments in bold and our responses in normal format immediately below each comment. Where necessary, page and line numbers in the responses are given to indicate the position of text in the revised manuscript.

Kind regards on behalf of all authors,
Cas Eikenaar

Reviewer: 1

Comments to the Author(s)

This manuscript presents very interesting data on recovery and activation of the immune system in a migratory species on stopover. They have addressed well the previous reviewer's comments and their manuscript has improved even more as a result. I only had very minor observations that authors can address easily.

Thank you for your supportive words.

L44 – Why (novel) in parenthesis?

We agree that the parentheses are not necessary and removed these.

L70, 77, 85, 177 – Perhaps these lines should be the start of new paragraphs? It might be that something happened when formatting the proofs, but both the introduction and the discussion seem to be one single huge paragraph. Please check and make sure that the text is appropriately formatted with paragraphs.

We agree that for readability, it is better to start a new paragraph at line 70, however, we do not think new paragraphs are appropriate at the other indicated positions (there is no switch in topic there and a new paragraph we believe would break the flow).

L76 – Please check whether the expression 'birds at stopover' should be changed to 'birds on stopover'. This is repeated in various sections throughout the manuscript and I believe that most literature uses 'on stopover'. I may be wrong.

We are not sure whether one of the two is preferable, but the first author (CE) has always used "at stopover" in his previous papers. Therefore we would like to keep this wording.

L161 – Please add a short sentence saying that you could not test for age effects. Supplementary data - Please add a column for sex and one for age in the supplementary material.

We have done so: "The limited number of adult birds in our sample precluded testing for an age effect on the changes in CIF." line 169-170

Sex and age have been added to the supplementary material, thank you for pointing this out.

Reviewer: 2

Comments to the Author(s)

This study describes the ability of a long-distance migrant to rapidly increase constitutive innate immunity during a stopover with ad libitum food. The study is well written and clearly laid out. It adds to our knowledge of the role that stopovers play in the ecology and physiology

of migrating birds as well as emphasising how immune function is a dynamic process that can change over quite short time periods.

I have been asked to review this manuscript for a second time (previously for *Biology Letters*), so I only have minor comments that relate to improving the clarity, and perhaps making use of the fact that the longer format of *RSOS* compared to *Biology Letters* allows for a more comprehensive presentation and discussion of the results. Thank you for addressing so thoroughly all the comments and suggestions that I made in my previous review of your manuscript. I hope you find these additional comments useful.

Thank you for your positive words and useful comments.

L37 – Change to “Here, we show that...”

Done.

L74-L77 – I was confused by the terminology used here. Constitutive immunity often is inflammatory, so this doesn't quite seem to make sense. Perhaps, within the innate immune system you are making the distinction between constitutive innate immunity and induced innate defences (see Lee 2006 Linking immune defenses and life history at the levels of the individual and the species. *Integ. Comp. Biol.* 46: 1000-1015)? If so, this needs to be made clear.

We agree that an inflammatory immune response is part of the innate immune system. However, with constitutive immune function, we refer to the baseline levels of innate immune function in the absence of an infection. This standing line of innate immune function does not include an inflammatory immune response when a pathogen starts replicating in the body, because an immune response is differently regulated and has different costs and benefits compared to baseline (constitutive) immune function. In particular the immune response includes typical sickness responses such as fever, increased body temperature, lethargy and anorexia, while those are not part of constitutive immune function. Hence it is important to separate those two. We have clarified this in the revision:

“An animal's immune system includes both constitutive (baseline) defences, the always present immediate first lines of defence, and induced immune responses [5,6]. The main benefit of an improved constitutive immune function is probably increased likelihood of clearing pathogens before they start replicating and establishing themselves inside a host, while an innate immune response is mounted when a pathogen starts replicating and establishing itself in the body. Baseline immune function and immune responses are differently regulated and baseline immune function often does not correlate with the strength of an immune response [7,8]. While immune responses are important to ensure survival, immune responses are energetically costly [9,10], always include self-damage, and often include “opportunity costs”, i.e. lost opportunities in life-history, resulting in fitness costs [10-13]. Hence, a stronger constitutive (baseline) immune function (CIF) should be favoured in order to avoid mounting an actual immune response.” lines 71-81.

At L76 you say that stopover migrants might prefer to use CIF because they are keen on avoiding immune responses, but CIF is a form of immune response/defence. I think you mean that stopover migrants might not want to use induced or adaptive immune responses, because these take longer to develop, and in the case of induced innate immune defences, are energetically costly?

CIF indeed is a form of immune defense, but it does not include an immune response. The birds can thus use CIF to prevent having to mount an immune response. Also see our response above.

L78 – The context of racing with pigeons might be a bit lost to the reader here, unless they are familiar with the paper. Consider changing to something like “...although intensive flight during races does not affect CIF in non-migratory racing pigeons...”

We agree and thank you for this suggestion, which we have followed.

L92 – Consider changing to “...can negatively affect immune defences.”

A good idea, the word 'negatively' has been inserted

L96 – Perhaps consider adding in a sentence here to make it clear how long you held birds for. Otherwise it feels a little like you dive straight into the methods. Something like “We captured migrating wheatears at stopover and held them for three days in captivity (which mimics the average stop-over time of this species at our study site), to assess whether they could boost CIF during this time. Each bird was blood-sampled...” Currently the text reads a bit like you are still in Biology Letters format, with minimal space for providing more thorough details. I think you can afford to include a few more words now. I also think it is worth to state upfront that the three days in captivity matches a typical stop-over period for this species. I realise that you do this at L187-190 (and should continue to do so – thanks for making that change from the previous version), but I think including it in brief detail early on helps reassure the reader that your study has direct biological relevance to stopover behaviour in this species.

This is a good point and we agree that it helps the reader to already in the Introduction better understand the design of our study. Thank you! We have added the following sentence: “We captured wheatears at stopover and held them in captivity for three days (matching stopover durations of this species [25,26]), and assessed whether they can boost CIF during this time.” Lines 102-104

L101 – I think it would be beneficial to finish with some predictions here, just to remind the reader what it is you are aiming to test with your study. Something along the lines of “If migratory wheatears are able to boost CIF during stop-over, then this predicts that CIF should be higher at day three than at day one, whereas if wheatears are unable to boost CIF, we would expect no difference in levels between the start and end of captivity.”

We normally are quite keen on predictions at the end of the Introduction, but in this instance we really did not know what to expect when we started the study and we do not want to make predictions afterwards as this is unethical. Also, with the addition of the sentence containing “...and assessed whether they can boost CIF during this time” (see comment and response above), we think it is clear what we are testing.

L106 – Change to “After capture, birds...”

Done.

L124-126 – Thanks for now including this clear explanation.

You are welcome! It helped improving the manuscript.

L169 – Change to “...to substantiate this hypothesis have, until now, been lacking.”

Done.

L171-172 – Again, I would encourage greater clarity in terminology here. What do you mean by “better CIF and immune responses”? CIF is a type of immune response.

No, CIF is not a type of immune response, please also see our responses to the very similar comments above. Do note that we have changed the ‘better CIF’ to ‘stronger CIF’, as that is more appropriate.

L181-182 – This point about encountering novel pathogens seems particularly relevant to young birds that have not previously migrated. Given that your sample included mostly first year birds, do you think you would have found similar results if you had only captured adult birds?

Yes, we do think so as passerine migrants are not stopover-site faithful, and thus may encounter novel pathogens during migrations as adults too.

Your sample sizes may not permit it, but I wondered if it might be informative to test for an age effect (first year vs adult birds) on change in immune parameters? Perhaps older birds are more experienced which might affect their flight performance and thus physical state when captured, which could in turn affect ability to upregulate CIF? The fact that you found no effect of food intake or changes in fuel stores would suggest this is not the case, but it could be

interesting to check, just in case.

Indeed, with only 8 adult birds, our sample size does not allow for a proper test of an age effect. We now point this out in the revised manuscript: “The limited number of adult birds in our sample precluded testing for an age effect on the changes in CIF.” line 169-170

(FYI, when eye-balling the data, there is no hint whatsoever for a potential age effect.)

Figure 1 – This figure really clearly illustrates your results. One striking thing is the result for microbial killing capacity. You can almost completely split the data here into two groups. Those birds that had 100% killing ability at day 1 maintained it at day 3. Perhaps these individuals can be considered as ‘high quality’ in some way? For most other individuals, killing capacity at day 1 was very low – less than 20%. Yet at day 3 it had shot up to greater than 80%. This seems like a dramatic change, and worthy of more discussion, especially when contrasted with the results for lysis and agglutination, where there was much more variation. Is there something specific about microbial killing capacity that makes it more important to up-regulate than lysis or agglutination? Or is this a cheaper form of CIF, so can be more easily up-regulated? Given that you are now not word count-limited, you might consider elaborating on your results a bit more in the discussion.

We agree that this is an interesting pattern. The reason for a stronger increase in BKA compared to lysis and agglutination could be that the bacteria killing capacity is a broader, more integrative measure compared to the other two parameters. With other words, while complement activity (lysis titer) and natural antibody titers (agglutination titer) reflect single parameters, BKA captures several sub-parameters of innate immune function simultaneously and the data suggest that some parameters can be upregulated faster than complement activity and natural antibodies titers respectively. This could include for example certain proteins or white blood cell types. We have included a short discussion in the revision:

“Interestingly, for many individuals the increase in BKA appears stronger than the increase in lysis. This is probably because the bacteria killing capacity is a broad, integrative measure of innate immune function [32,53,54]. In contrast, complement activity (lysis titer) and natural antibody titers (agglutination titer) reflect more tailored parameters [34, and references therein]. Our data suggest that some parameters (captured in the BKA) might be upregulated to a stronger degree in short time frames than complement activity and natural antibodies titers respectively.” lines 194-200

Supplementary data – Please include sex and age data in your Excel file.

Sex and age have been added to the supplementary material, thank you for pointing this out.